# APPROXIMATE NEAREST NEIGHBOR SEARCH THROUGH MODERN ERROR-CORRECTING CODES

**Noam Touitou**
Amazon
noamtwx@gmail.com

**Nissim Halabi**
nissimh@gmail.com

## ABSTRACT

A locality-sensitive hash (or LSH) is a function that can efficiently map dataset points into a latent space while preserving pairwise distances. Such LSH functions have been used in approximate nearest-neighbor search (ANNS) in the following classic way, which we call classic hash clustering (CHC): first, the dataset points are hashed into a low-dimensional binary space using the LSH function; then, the points are clustered by these hash values. Upon receiving a query, its nearest neighbors are sought within its hash-cluster and nearby hash-clusters (i.e., multi-probe). However, CHC mandates a low-dimensional latent space for the LSH function, which distorts distances from the (high-dimensional) original real space; this results in inferior recall. This is often mitigated through using multiple hash tables at additional storage and memory costs.

In this paper, we introduce a better way of using LSH functions for ANNS. Our method, called the Polar Code Nearest-Neighbor (PCNN) algorithm, uses modern error-correcting codes (specifically polar codes) to maintain a manageable number of clusters inside a high-dimensional latent space. Allowing the LSH function to embed into this high-dimensional latent space results in higher recall, as the embedding faithfully captures distances in the original space. The crux of PCNN is using polar codes for probing: we present a multi-probe scheme for PCNN which uses efficient list-decoding methods for polar codes, with time complexity independent of the dataset size. Fixing the choice of LSH, experiment results demonstrate significant performance gains of PCNN over CHC; in particular, PCNN with a single table outperforms CHC with multiple tables, obviating the need for large memory and storage.

## 1 INTRODUCTION

In similarity search, one is first given a dataset $D$ of points, then a set of query points from the same space. For each query, the goal is to find the closest point (or points) in $D$ to that query, according to some given metric. The simplest way to find these nearest neighbors of a query is to calculate the distance of the query from each point in the dataset $D$; however, when the dataset $D$ is large, this linear cost in the dataset's size is prohibitive. Thus, upon query, one would like to consider only a small subset of $D$. Since these non-exhaustive algorithms do not consider all points in the dataset, we are interested in *approximate* similarity search, with the following relaxations:

- We allow an approximation ratio, i.e., the algorithm is allowed to return neighbors whose distance to the query is at most some factor $\alpha \geq 1$ times the distance of the nearest neighbor to the query[1].
- Since the algorithm does not explore all points, it can sometimes return a result which is not within the desired distance; the fraction of results which are within the desired range is called the *recall* of the algorithm.

**Clustering Methods.** A common technique for approximate similarity search is to divide the dataset $D$ into clusters. Then, upon receiving a query, the algorithm would only search the points of $D$ that appear in the clusters which are closest to the query. (If the clusters are represented by points in the original space, the distance to the query is well defined. Otherwise, a different metric is needed.) Algorithms based on such clustering are often used in practice, since they allow storing different clusters on different machines (i.e., sharding) for efficient distributed processing of queries.

---

[1]For similarity measures, which should be maximized, we would instead be interested in $\alpha \leq 1$.

In light of these benefits of clustering methods, we would like to study them in our approximate setting. However, existing clustering methods have some drawbacks in this setting, which motivate the algorithm presented in this paper. We now explore two popular clustering methods and describe their drawbacks.

**Clustering by Training Cluster Centers.** In this method, introduced by Sivic & Zisserman (2003), cluster centers are trained on the dataset (or some sample of the dataset) using a clustering algorithm (namely k-means), and each dataset point is mapped to a cluster (e.g., by the nearest cluster center). Upon query, the clusters corresponding to the closest cluster centers are searched. This common algorithm is part of the popular Faiss library (Johnson et al., 2021) as the default inverted-file (IVF) method; we henceforth refer to this method as IVF. Since the cluster centers are unstructured, finding these closest centers requires a number of distance computations which is linear in the number of centers. The total number of distance computations (for both centers and dataset points) is therefore always at least the square root of the dataset's size. Thus, while popular in general, this method is less appropriate for an approximate regime in a large dataset, as we would like to get high recall using a much smaller computational cost, independent of the dataset size.

**Clustering by Locality-Sensitive Hashing.** Another such clustering method uses Locality-Sensitive Hashing, or LSH (Indyk & Motwani, 1998). In this method, a locality-sensitive hash $h : \mathbb{R}^d \to \{0, 1\}^{\mathsf{nbit}}$ is used to map the dataset to hash codes (nbit-bit strings). Each such hash-code identifies a cluster which contains the dataset points in the hash-code's preimage. Upon receiving a query $q$, the hashcode $h(q)$ is calculated, and closest points are search only within the clusters identified with the closest hashcodes to $h(q)$ (Lv et al., 2007). We refer to this simple method, which is used in most classic LSH papers, as Classic Hash Clustering, or CHC for short. Note that the choice of the LSH function $h$ is independent from the operation of CHC, and should be chosen such that distances in the embedding space approximate distances in the original space; two possible choices (which we also consider in this paper) are hyperplane LSH (Charikar, 2002) and the data-dependent autoencoder LSH (Tissier et al., 2019).

The fact that CHC uses LSH functions provides some advantages. First, the closest clusters to a query can be found without calculating its distance to *every* cluster; this makes CHC more suitable for the approximate regime than IVF. Second, the index in CHC can be easily augmented with additional dataset points in an online fashion, as the clustering is not trained on the dataset. In addition, CHC usually has a low memory/storage footprint. However, using CHC does not usually achieve high recall; this is usually alleviated by using multiple tables (i.e., multiple clusterings), at significant memory and storage costs.

Why does CHC achieve low recall? A possible explanation could be that the distances between the dataset/query points in high-dimensional space $\mathbb{R}^d$ are not faithfully captured by the hashing to the space $\{0, 1\}^{\mathsf{nbit}}$. This is since the hash-code space must be low-dimensional, as memory and running time restrictions make it infeasible to use large nbit. For example, if one chooses $\mathsf{nbit} = 40$ (and thus $2^{40}$ clusters) for a dataset of a billion points, 99.9% of the clusters would be empty; Thus, nbit must remain small, and usually does not exceed 32. However, this severely limits the granularity of distances, as Hamming distances in this low-dimensional binary space only take on one of 32 non-zero values. This lack of granularity is a property of every low-dimensional embedding, and thus appears in *all* LSH functions (including data-dependent functions). In addition, using a low number of embedding bits could yield a high variance in the distance of any embedded pair of points. For example, consider hyperplane LSH: in this method, the expected relative Hamming distance of the hash-codes is equal to the relative angular distance between the original points (Charikar, 2002), but the bits of the hashcode are generated independently. Thus, the deviations from the expectation are very significant when the number of bits is low, as mandated by CHC. These drawbacks of CHC thus call for a different technique, which is able to simultaneously utilize distance information from a high-dimensional binary embedding, as well as preserve a reasonable number of clusters.

**Our Contributions.** In this paper, we present a generalization of CHC which uses modern error-correcting codes (ECCs); we call this method the Polar Code Nearest Neighbor algorithm (or *PCNN*). PCNN encapsulates any LSH method $H$, similar to CHC, but yields superior performance. CHC uses $H$ to embed into a nbit-dimensional binary space for some low nbit (e.g., $\mathsf{nbit} = 30$); PCNN instead uses $H$ to embed into a cdim-dimensional binary space, for some $\mathsf{cdim} \gg \mathsf{nbit}$ (e.g., $\mathsf{cdim} = 512$). Then, the embedded dataset in PCNN is clustered, such that the set of clusters forms a nbit-dimensional subspace inside the larger cdim-dimensional embedding space. Upon query, the

probing procedure is performed in the high-dimensional embedding space. As we later discuss, CHC is a special case of PCNN in which $\mathsf{nbit} = \mathsf{cdim}$.

By separating the dimension of the embedding space $\mathsf{cdim}$ from the dimension of clusters $\mathsf{nbit}$, PCNN addresses the previously-discussed shortcoming of CHC, i.e., the low dimensionality of the resulting embedding which leads to a distortion of distances. PCNN performs probing on a large, $\mathsf{cdim}$-dimensional space, in which distances between embedded points better approximate the distances in the original space. At the same time, PCNN maintains the same number of clusters as CHC (i.e., $2^{\mathsf{nbit}}$). In addition, PCNN enjoys the benefits of CHC: it has an index which is small and easily extensible, as well as an efficient probing method whose running time does not depend on the number of clusters.

*Polar codes and List Decoding.* The crux of our algorithm is the choice of cluster centers in this high-dimensional binary space: these centers are chosen to allow efficient mapping from a binary point to the closest centers, for the sake of multi-probe. This is where we use recent advances in error-correcting codes, namely the modern *polar codes*: choosing the centers to be the codewords of a polar code allows us to use *list-decoding*, an ECC technique which efficiently maps from a binary word to the closest $\mathsf{nprb}$ codewords, for any parameter $\mathsf{nprb}$. Specifically, list decoding to the $\mathsf{nprb}$ closest codewords (i.e., multi-probe in PCNN to find the $\mathsf{nprb}$ closest clusters) runs in time $O(\mathsf{nprb} \cdot \mathsf{cdim} \log \mathsf{nbit})$; this is nearly optimal, as $\mathsf{nprb} \cdot \mathsf{cdim}$ is the representation size in bits of the $\mathsf{nprb}$ closest cluster centers themselves.[2] (See Appendix D for detailed complexity comparison.)

*Evaluation.* We evaluate PCNN empirically on real-valued real-world datasets, and establish that it performs significantly better than standard (multi-probe) CHC. As PCNN can be used to encapsulate any LSH method, we chose to evaluate PCNN against CHC on two very different LSH methods: the first is the classic hyperplane method (Charikar, 2002), and the second is a data-dependent method based on the output of an autoencoder (Tissier et al., 2019). Moreover, we also show that PCNN outperforms CHC *with multiple tables*, while having a memory and storage footprint identical to that of single-table CHC. This implies that PCNN is a strong alternative to using CHC with multiple tables.

We also evaluate PCNN on binary datasets, where both PCNN and CHC run directly on the dataset points (and thus a real-to-binary LSH is not needed). The baseline here is provided by the IndexBinaryMultiHash class from Faiss (Johnson et al., 2021). The results mirror those for real datasets, showing a clear advantage to PCNN.

In summary, in this paper, we give the following contributions:

- We present the PCNN algorithm as a new clustering method for approximate nearest-neighbor search. The PCNN algorithm uses error-correcting codes – specifically polar codes – to index according to a high-dimensional binary embedding while keeping the number of clusters low.
- We provide a multi-probe scheme for PCNN, which is based on efficient list-decoding algorithms for polar codes.
- We evaluate PCNN vs. multi-probe CHC as a baseline, and show robust performance gains.

Source code of the PCNN algorithm and the evaluations presented in the paper can be found on https://github.com/amzn/amazon-nearest-neighbor-through-ecc.

## 2 RELATED WORK

ANNS is fundamental to applications in many domains with various specifications' trade-offs, including preprocess and search time complexity, search quality, memory size, scalability with dataset size and data dimension, robustness to query workloads and dataset updatability, and more (Li et al., 2020; Aumüller et al., 2020). Two main categories of ANNS are graph-based methods and inverted index clustering based methods. Graph-based algorithms such as (Hierarchical) Navigable Small World graphs (Malkov et al., 2012; 2014; Malkov & Yashunin, 2020), NSG (Fu et al., 2019), and DiskANN (Jayaram Subramanya et al., 2019) achieve good performance in a non-distributed setting.

This paper focuses on clustering methods for ANNS, in which their support for sharding makes them commonly used in practice for billion scale updatable datasets with high query workloads. As outlined in Section 1, clustering methods for ANNS may be divided into two popular types: trainable

---

[2]As noted later in the paper, a good choice for $\mathsf{cdim}$ is the original real dimension $d$. In this regime, the cost of the binary embedding $\Theta(d\mathsf{cdim}) = \Theta(\mathsf{cdim}^2)$ dominates the cost of the list decoding.

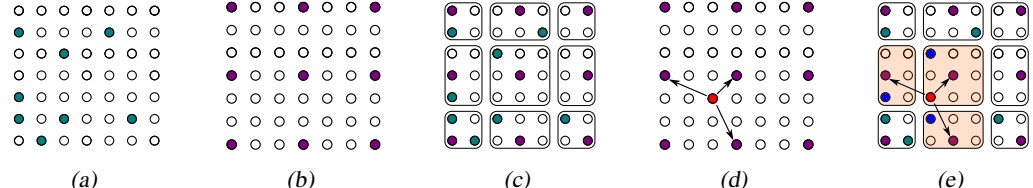

*Figure 1: A visual illustration of the PCNN algorithm (Algorithms 1 to 3) using the* cdim*-dimensional binary cube sketched by a grid of $2^{cdim}$ points. Sub-figures (a)-(c) deal with initialization and preprocessing and sub-figures (d)-(e) deal with querying. (a) Embedding of the dataset vectors in $D$ by $h : \mathbb{R}^d \rightarrow \{0,1\}^{cdim}$ (green solid). (b) $2^{nbit}$ polar $[$cdim, nbit$]$-code codewords $C \subseteq \{0,1\}^{cdim}$ (purple solid) whose* nbit*-bits identifiers comprise the index entries. (c) Partition $M$ of the dataset $D$, comprised of $2^{nbit}$ partitions. The partitions correspond to Voronoi cells (rectangles) induced by the codewords in $C$ and computed by* LISTDEC *with $\ell = 1$. (d) Extracted* nprb $= 3$ *cluster IDs of focus for a query $q$. Retrieved by* LISTDEC *of $h(q)$ (red solid) with $\ell =$* nprb*, using time proportional to* nprb *that does not depend on $|C|$. (e) Find the nearest vectors to $q$ with respect to the subset of $D$ contained in the extracted* nprb *clusters of focus.*

(data dependent) and structured (data independent). The main line of works for trainable clustering methods follows from Sivic & Zisserman (2003) through the popular Faiss library (Johnson et al., 2021) to state-of-the-art algorithms such as SPANN (Chen et al., 2021). Yet, the time complexity of these algorithms remains dependent in the dataset size (at least square-root in the case of Faiss/IVF and poly-logarithmic in the case of SPANN). Modern machine learning techniques (such as neural networks) are also studied lately (e.g., (Kraska et al., 2018; Wang et al., 2018; Dong et al., 2020)), dealing with space partitioning of the dataset. However, the need to search for closest clusters (for multi-probe) still remains dependent in the dataset size.

Locality-sensitive hash functions have seen much previous work; see for example Wang et al. (2014); Jafari et al. (2021); Charikar (2002); Andoni et al. (2015); Terasawa & Tanaka (2007); Laarhoven (2017). Often, such LSH functions are used for clustering using CHC (and possibly using multi-probe or multiple tables). Another popular use is binarization for efficient distance computations (e.g., for speeding up exhaustive search). Notably, recent works on LSH introduced sparse high-dimensional hash codes for similarity search inspired by the fly's olfactory circuit Dasgupta et al. (2017); Sharma & Navlakha (2018); Ryali et al. (2020). However, these hash codes lack a multi-probe technique in high dimensions which is necessary for high recall, and result to recursively reduce to low dimensions for multi-probe.

## 3   THE PCNN ALGORITHM

The main idea of the PCNN algorithm is to embed the real-valued dataset $D \subseteq \mathbb{R}^d$ from the original real-valued space into a high-dimensional binary space $\{0,1\}^{cdim}$, but only allow some much-smaller subset $C \subseteq \{0,1\}^{cdim}$ to be cluster identifiers, where $|C| = 2^{nbit}$ for nbit $\ll$ cdim. The index comprises the cluster identifiers of all points in the dataset. Upon receiving a query $q \in \mathbb{R}^d$, the algorithm would embed $q$ into $\{0,1\}^{cdim}$, find the closest cluster identifiers in $C$, and search within those clusters. In such a setting, cdim would be chosen as large enough to faithfully capture distances in the original real space (as discussed in the introduction for, e.g., hyperplane LSH (Charikar, 2002)), while $|C|$ would be chosen through memory and running-time considerations. For example, for a dataset $D \subseteq \mathbb{R}^{128}$ such that $|D| = 2^{24}$, one could choose cdim $\simeq 128$ and nbit $= 24$.

In implementing this algorithm, a technical problem arises: upon receiving a query, how does one efficiently find the closest clusters in $C$ to that query? We solve this problem using error-correcting codes.

**Error-Correcting Codes (ECCs) and Polar Codes.** A linear, error-correcting $[N, K]$-code is a subspace $C$ of dimension $K$ in $\{0,1\}^N$. By and large, a good code $C$ should satisfy two properties. First, the words in $C$ (called the codewords) should be spaced, such that the distance between any two such words is large. Second, the code should have an algorithm for efficient *decoding*, i.e.,

mapping from an arbitrary word in $\{0,1\}^N$ to the closest codeword in $C$ (in Hamming distance). Another, more advanced property is efficient *list decoding*, which is mapping from such a word to the closest nprb codewords in $C$, for some parameter nprb. Note that good codes require careful design. For example, random linear codes have good distance properties, but do not admit efficient decoding algorithms (specifically, decoding such codes is NP hard (Berlekamp et al., 1978)). Unlike classical error-correcting codes, modern error-correcting codes have a structure that mimics random linear codes, but admit efficient probabilistic decoding (Richardson & Urbanke, 2008).

A recent family of such modern codes is *polar codes*, introduced by Arikan (2009). These codes support every choice of $N$ and $K$, which allows for added flexibility. In addition, Tal & Vardy (2015) gave an efficient *list-decoding* algorithm for polar codes; this is useful for designing a multi-probe scheme for our algorithm. Thanks to their performance guarantees and ease of implementation, these codes have become prevalent in recent years (e.g., as part of the 5G cellular standards), and efficient implementations for polar codes exist in both hardware and software (Cassagne et al., 2019a;b). These properties prompted us to choose polar codes for similarity search. More details about polar codes and the (list-)decoding process can be found in Appendix A.

**The PCNN Algorithm.** To summarize, we describe the PCNN algorithm on a $d$-dimensional dataset $D$ of $n$ points. This algorithm is parameterized by the parameters cdim and nbit, as well as nprb (the number of clusters to probe upon query). The PCNN algorithm contains 2 main parts: (i) Initialization and preprocessing (Algorithm 1) for generating the index of the PCNN clustering method, and (ii) querying for the sizenn nearest neighbors to $q$ within nprb clusters of focus (Algorithm 2). Figure 1 provides a visual illustration of the PCNN algorithm; for presentation purposes we depict the cdim-dimensional binary cube as a grid of $2^{\mathsf{cdim}}$ points.

During initialization (see Algorithm 1), a code mask $\mathbf{r} \in \{0,1\}^{\mathsf{cdim}}$ is generated, where $\|\mathbf{r}\|_1 = \mathsf{nbit}$, which represents the polar $[\mathsf{cdim}, \mathsf{nbit}]$-code to be used in the algorithm. This mask is generated through a genie-aided process (Arikan, 2009). During the preprocessing of the dataset, we cluster the dataset points according to the nearest codeword to their binary embedding. Specifically, an empty partition $M$ of the dataset points into $2^{\mathsf{nbit}}$ clusters is created (where each cluster identifier is a nbit-bit string). Then, each dataset point is embedded into cdim-dimensional binary space using hyperplane LSH $h : \mathbb{R}^d \to \{0,1\}^{\mathsf{cdim}}$ (or any other LSH function). Next, the function LISTDEC (see Algorithm 3) is called on the binary dataset point with the argument $\ell = 1$, to obtain the single closest codeword $\mathbf{c}$ to that binary point. (LISTDEC runs list decoding with slightly-larger list size $\ell' \geq \ell$, then takes the closest $\ell$ codewords; see Appendix A for more details.) Since there are only $2^{\mathsf{nbit}}$ codewords, the algorithm extracts a nbit-bit cluster identifier $\mathbf{a}$ for $\mathbf{c}$; this identifier is a subset of bits from $\mathbf{c}$, namely those bits whose indices get a value of 1 in the mask $\mathbf{r}$ (this identifier is unique; see Appendix A). The dataset point is then added to the cluster $M_{\mathbf{a}}$ in the partition $M$. The total running time of this preprocessing procedure is thus $n \cdot (O(\text{embedding cost}) + O(\mathsf{cdim} \cdot \log \mathsf{nbit}))$.

Upon receiving a query (Algorithm 2), the algorithm first retrieves nprb cluster IDs of focus, and then searches within for the sizenn nearest neighbors to $q$. Algorithm 2 embeds the query into cdim-dimensional binary space (using the same embedding $h$ as used in Algorithm 1), then calls the function LISTDEC with the argument $\ell = \mathsf{nprb}$ to obtain the nprb closest codewords $\mathbf{c}^0, \cdots, \mathbf{c}^{\mathsf{nprb}-1}$ to the binary dataset point. The cluster identifiers of these points are then extracted as before, which yields nprb clusters of focus in $M$ to be probed. The algorithm then goes over all points in the chosen nprb clusters, calculates the distance of the query to each such dataset point in a chosen cluster, and finds the closest sizenn points to the query. These distance comparisons take place in the original real space for maximum accuracy (although quantization methods, such as product quantization, could be applied as well). The running time for a query is thus $O(\text{embedding cost}) + O(\mathsf{nprb} \cdot \mathsf{cdim} \cdot \log \mathsf{nbit})$ (plus, of course, the cost of the distance comparisons within each cluster).

Figure 2 visualizes the process of extracting $\mathsf{nprb} = 4$ cluster IDs of $\mathsf{nbit} = 8$ bits from a query in $\mathbb{R}^{10}$. The query is embedded to a binary space with $\mathsf{cdim} = 16$, list-decoding with $\ell = \mathsf{nprb} = 4$ is applied on the embedded vector, where cluster IDs are extracted according to a code mask $\mathbf{r} = 0000001100111111$ (indicated by the red bits).

**PCNN as a Generalization of CHC.** Note that PCNN is in fact a generalization of CHC: choosing $\mathsf{cdim} = \mathsf{nbit}$ would imply $C = \{0,1\}^{\mathsf{cdim}}$, i.e., every word is a codeword. In addition, the cluster

**Algorithm 1** PCNN: Initialization and Preprocessing

*Parameters:*
nbit – the dimension of the binary cluster identifier.
cdim – the length of a codeword.
nprb – the number of clusters to probe upon query.
INITIALIZATION:

Generate a polar code mask $\mathbf{r} \in \{0,1\}^{\mathsf{cdim}}$ with exactly nbit 1-valued entries.
Generate a binary LSH embedding $h : \mathbb{R}^d \to \{0,1\}^{\mathsf{cdim}}$.
Initialize the index $M = (M_\mathbf{a})_{\mathbf{a} \in \{0,1\}^{\mathsf{nbit}}}$ to contain empty sets.

PREPROCESSING($D$):

**for** $\mathbf{p} \in D$ **do**
    Let $\mathbf{b} \leftarrow h(\mathbf{p})$
    {get the closest codeword to $\mathbf{b}$.}
    Let $\mathbf{c} = (c_0, \cdots, c_{\mathsf{cdim}-1}) \leftarrow$ LISTDEC($\mathbf{b}, 1$).
    {extract the nbit-bit cluster identifier for $\mathbf{c}$.}
    Let $\mathbf{a} \leftarrow (c_i)_{i|r_i=1}$.
    {add the dataset point to the cluster.}
    Add $\mathbf{p}$ to $M_\mathbf{a}$
**end for**

**Algorithm 2** PCNN: Upon Query

UPONQUERY($\mathbf{q}, \mathsf{sizenn}$):

Let $\mathbf{b} \leftarrow h(\mathbf{q})$.
{get the closest nprb codewords to $\mathbf{b}$.}
Let $\mathbf{c}^0, \ldots, \mathbf{c}^{\mathsf{nprb}-1} \leftarrow$ LISTDEC($\mathbf{b}, \mathsf{nprb}$).
Let $S$ be an empty list of (distance, point) pairs.
**for** $i \in \{0, \cdots, \mathsf{nprb}-1\}$ **do**
    Denote $\mathbf{c}^i = (c_0^i, \cdots c_{\mathsf{cdim}-1}^i)$.
    {extract the nbit-bit cluster identifier for $\mathbf{c}^i$.}
    Let $\mathbf{a} \leftarrow (c_j^i)_{j|r_j=1}$.
    **for** $\mathbf{p} \in M_\mathbf{a}$ **do**
        Set $\delta_\mathbf{p} \leftarrow d(\mathbf{q}, \mathbf{p})$.
        Add $(\delta_\mathbf{p}, \mathbf{p})$ to $S$.
    **end for**
**end for**
Choose and return the sizenn entries in $S$ with the smallest distance.

**Algorithm 3** PCNN: List Decoding Wrapper

LISTDEC($\mathbf{b}, \ell$):

{ This function returns the $\ell$ closest codewords in $C$ to $\mathbf{b} \in \{0,1\}^{\mathsf{cdim}}$ (with high probability). For more details, see Appendix A.}
Define $\ell' \leftarrow f(\ell) \geq \ell$, for $f$ as defined in Appendix A.
Use the list-decoding algorithm of Hashemi et al. (2016) on $\mathbf{b}$ with list size $\ell'$ to obtain a set $S$ of codewords, where $|S| = \ell'$.
Return the $\ell$ nearest codewords to $\mathbf{b}$ in $S$.

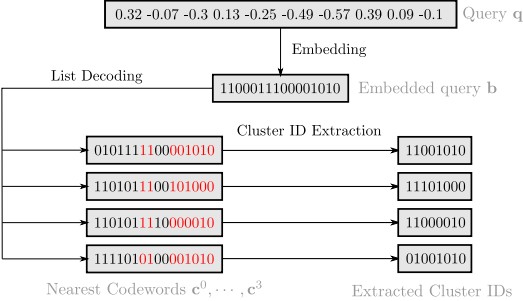

*Figure 2: Obtaining cluster IDs by PCNN upon query.*

ID of a codeword would be the entire codeword, and the closest codewords to a binary-embedded query $q$ would exactly be those words generated by bit flips in the multi-probe scheme of CHC.

## 4 EXPERIMENTS

In this section, we empirically evaluate the PCNN algorithm.

### 4.1 DATASETS

We consider three representative real-world datasets for evaluation as follows.[3]
1. *BIGANN*: SIFT descriptors applied to images (Dataset, 2010).

---

[3] The datasets and queries are taken from the Billion-Scale Approximate Nearest Neighbor Search Challenge in NeurIPS'21 (Competition, 2021)

| Dataset Name | #Dataset/#Query | Dim. | Distance | Normalized |
|---|---|---|---|---|
| *YandexTTI* | 10M/5K | 200 | Cosine | Yes |
| *YandexDeep* | 10M/5K | 96 | Euclidean | Yes |
| *BIGANN* | 10M/5K | 128 | Euclidean | No |

*Table 1: Evaluation datasets: main characteristics.*

2. *Yandex-Deep1B (YandexDeep)*: image descriptor dataset consisting of the projected and normalized outputs from the last fully-connected layer of the GoogLeNet model (Babenko & Lempitsky, 2016), which was pretrained on the Imagenet classification task (Babenko & Lempitsky, 2016).

3. *Yandex Text-to-Image-1B (YandexTTI)*: A cross-model dataset (text and visual) where the dataset consists of image embeddings and the queries are textual embeddings (Dataset, 2021).

Table 1 summarizes the main characteristics of each dataset. All datasets consist of 10M points, and use 5K queries for evaluation. For *YandexDeep* and *BIGANN* we use Euclidean distance, while for *YandexTTI* we use cosine distance (Appendix C explores the effect of approximate cosine distance on cosine similarity).

## 4.2 EVALUATION METRICS

Upon a query for the sizenn nearest neighbors of some point $q$, an algorithm's output $S$ consists of sizenn points in the dataset. Denote the distance function of the chosen dataset by $d(\cdot, \cdot)$, and define $\delta_i^q$ to be the distance of the $i$'th closest dataset point to $q$. For some approximation factor $\alpha \geq 1$, define the $\alpha$-*recall* of the algorithm to be $\left|\left\{x \in S \mid d(x, q) \leq \alpha \cdot \delta_{\text{sizenn}}^q\right\}\right| / |S|$. (When $\alpha$ is known, we sometimes refer to $\alpha$-recall simply as recall.)

The measure we use for the cost of a query is the number of distance calculations made by the algorithm, denoted by ndis. Indeed, since the algorithms we consider are clustering-based, their main cost is in comparing the query to the subset of the dataset in the chosen clusters; the number of distance calculations captures this cost. In this work, we therefore measure the cost and performance of an algorithm by the pair (ndis, recall). Varying the number of clusters probed by the algorithm, denoted nprb, controls this performance pair: increasing nprb would probe more points (increase ndis) but obtain better results (increase recall).

## 4.3 BASELINES

We compare PCNN to CHC for similarity search. Both clustering methods use an underlying LSH function; for our experiments, we use the classic hyperplane LSH, introduced by Charikar (2002), for both PCNN and CHC. Another choice of LSH is autoencoder LSH (Tissier et al., 2019); we consider this data-dependent LSH in Appendix B.6. For the (randomly-chosen) hyperplane LSH, we average over 30 different random seeds to reduce undue variance (see Appendix B.3 for more on this variance).

Note that in our setting of ANNS, high recall on the evaluated datasets that contain $10^7$ points is achieved by both PCNN and LSH using significantly less than $\sqrt{10^7}$ distance computations; thus, k-means-based clustering methods (such as IndexIVFFlat in Faiss (Johnson et al., 2021)) are not competitive in this regime, as they require at least $\sqrt{|D|}$ distance computations for a dataset $D$.

## 4.4 EXPERIMENTAL RESULTS

Having described the main ingredients of our experiments, we describe the experiments themselves. Due to space constraints we refer to Appendix B for further details on some experimental results outlined herein. The outline of the performance gains of PCNN over CHC is the following. In this section, we demonstrate the following results.

1. The performance of PCNN improves as cdim grows, thus outperforming CHC (which is the special case of PCNN in which nbit = cdim). The improvements plateau when cdim $\approx d$.

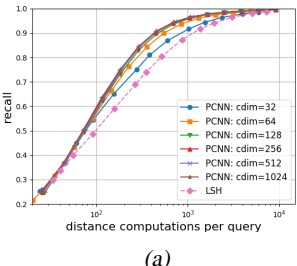 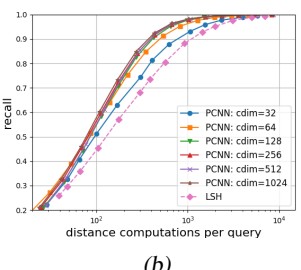 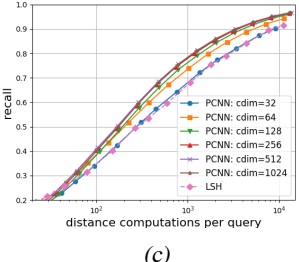

|  (a)  |  (b)  |  (c)  |

Figure 3: *recall/ndis* performance comparison for **sizenn** = 1 (using **nbit** = 28 and hyperplane LSH) for three datasets: (a) YandexDeep, $\alpha = 1.4$; (b) BIGANN, $\alpha = 1.4$; (c) YandexTTI, $\alpha = 2$.

2. PCNN with large **cdim** and a single table (**ntable** = 1) outperforms CHC with multiple tables (e.g., **ntable** = 8).
3. The performance gains of PCNN over CHC extend to binary datasets.

In addition, in Appendix B, we establish the following useful robustness properties.

1. The performance gains hold for multiple choices of approximation factor $\alpha$ and nearest neighbor size **sizenn**. (See Appendices B.1 and B.2.)
2. An efficient approximate hyperplane embedding, using random sign flips and Hadamard transform (Andoni et al., 2015), can be used for PCNN with negligible performance loss. (See Appendix B.5.)
3. The variance of PCNN performance curves upon different instantiations of binary embedding (by seeds) is lower than that of CHC. (See Appendix B.3.)
4. The superiority of PCNN over CHC remains when choosing the best trialed instantiation per operating point (i.e., performance of convex frontier). (See Appendix B.4.)
5. The performance gains of PCNN over CHC extend to different LSH functions, specifically a trainable autoencoder LSH of Tissier et al. (2019). (See Appendix B.6.)

**Performance Gains.** With the hyperplane LSH of Charikar (2002) as the underlying hashing method, we compared PCNN with various choices of **cdim** to CHC, over the three considered datasets; see Figure 3. Each curve in Figure 3 shows the average **recall/ndis** of an algorithm for various choices of **nprb**, and averaged over 30 different random seeds. In this experiment, we focus on **sizenn** = 1 (i.e., single nearest neighbor). For the approximation ratio, we chose $\alpha = 1.4$ for L2 datasets (*BIGANN*, *YandexDeep*) and $\alpha = 2$ for the cosine-distance dataset (*YandexTTI*). (Because cosine distance is proportional to L2 squared, these approximation ratios are roughly equivalent.) As we later discuss, similar results are also obtained for different choices of $\alpha$ and **sizenn**.

Figure 3 shows a marked improvement as **cdim** grows, which eventually plateaus at **cdim** = 128 for *BIGANN* and *YandexDeep*, and at **cdim** = 256 for *YandexTTI*. Since this seems to correspond to the real dimensionality of these datasets, we conjecture that choosing **cdim** to be roughly the dimension of the dataset is appropriate.

**Multi-Table LSH.** A common tool for increasing recall for CHC is using multiple tables, i.e., indexing the dataset according to **ntable** different binary embeddings of **nbit** bits, and probing clusters from all **ntable** resulting partitions upon query (Indyk & Motwani, 1998; Gionis et al., 1999). This method comes with additional costs over standard (single-table) CHC, notably its increased memory and disk usage, increased running time, and the need for deduplication.

In Figure 4a, we compare multi-table CHC to PCNN. Additionally, we consider *multi-table PCNN*, in which **ntable** different embeddings to **cdim** bits are used to create **ntable** tables (similar to CHC), to see if it offers improved performance over single-table PCNN. We consider the *YandexDeep* dataset, with **sizenn** = 1 and $\alpha = 1.4$ (as before). All algorithms use **nbit** = 28, and their performance is averaged over 30 different random seeds. It can be observed that single-table PCNN with **cdim** = 512 outperforms CHC with 8 tables; this is notable, as PCNN uses an index which is 8 times smaller. In addition, PCNN with **cdim** = 512 does not benefit from additional tables. The fact that CHC improves with additional tables and PCNN does not might imply that the only benefit of multi-table CHC is in using additional embedding bits; if this is the case, it could be supplanted by PCNN with a single table and large-enough **cdim**.

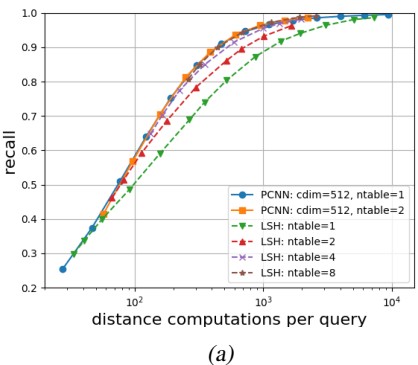
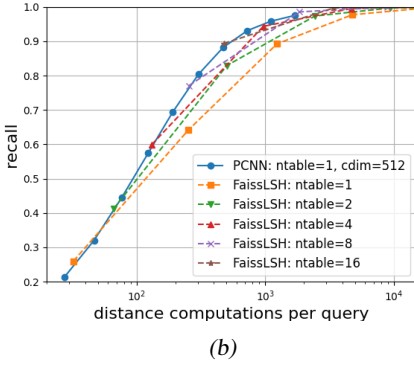

|     |     |
| --- | --- |
| *(a)* | *(b)* |

Figure 4: Comparison of multiple table usage for **sizenn** $= 1$ and $\alpha = 1.4$, using **nbit** $= 28$. *(a) On YandexDeep, using hyperplane LSH. (b) on a binary dataset obtained by embedding YandexDeep to binary space of dimension* $512$.

**Binary Datasets.** The PCNN algorithm can also operate on binary datasets (i.e., without need for binary embedding). To test PCNN on such a dataset, we use hyperplane embedding on the *YandexDeep* dataset to create a 512-dimensional binary dataset, then run PCNN and the baseline on this binary dataset. Note that this is different from the previous experiments, in which a real dataset was embedded into binary by PCNN/CHC; indeed, in this experiment the distance comparisons of the algorithms, as well as the ground truths for the queries, are all in the binary space. In binary datasets, the natural LSH technique is simply to take the first **nbit** bits of the dataset/query point as its hash-code; The IndexBinaryMultiHash (IBMH) class of Faiss (Johnson et al., 2021) implements CHC using this LSH technique. Thus, we use IBMH as our baseline for binary datasets. IBMH also supports multiple tables, through using the **ntable** · **nbit** first bits as hash-codes for the **ntable** tables.

For binary datasets, we again find that PCNN outperforms CHC: we replicate the results for real datasets and show that single-table PCNN performs as well as CHC with multiple tables. To mitigate any effects from the (deterministic) choice of hash-code bits by IBMH, we again average our results over 30 different random seeds; here, the seeds are used for creating the *dataset* rather than by the algorithms. It can be observed in Figure 4b that single-table PCNN performs as well as IBMH with 16 tables. (Note that the vertices in the curves representing IBMH are sparse, as the parameter controlling the number of probed clusters in IBMH is quite coarse.)

## 5    CONCLUSIONS

In this paper, we addressed a problem in clustering methods for similarity search. Choosing the cluster centers to be unstructured, as in $k$-means IVF, leads to high cost in finding clusters at query time. However, structured cluster centers, as used in CHC, are limited to low-dimensional embedded spaces, which distorts the metric space and hurts recall. We bridged the algorithmic gap in designing structured cluster centers in high-dimensional spaces using polar codes. These codes allow for a manageable number of clusters to exist in a high-dimensional space, and provide efficient multi-probe through list decoding. Indeed, the ample previous work done on these codes for more classic applications (e.g., forward error correction in communications) provides us with efficient list-decoding procedures, easily implementable in software or hardware. Through experiments, we've demonstrated the benefit of this high-dimensional embedding space, establishing (in particular) that CHC with multiple tables is superseded by PCNN with a single table, saving memory and storage.

For future work, various refinements and generalizations of PCNN for similarity search could be considered. For example, one could use polar codes with different granularity (controlled by the parameter **nbit** while preserving the same **cdim**), such that areas in the embedded space which are dense with the dataset would have a finer clustering. This refinement is motivated by mimicking unstructured clustering by hierarchies of structured clustering methods (Wang et al., 2018).

## ACKNOWLEDGEMENTS

We would like to thank Iftah Gamzu, Marina Haikin, Gal Levi, Alexander Lorbert and Uri Sharir for helpful discussions and feedback that helped improve the paper.

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

## A  POLAR CODES AND LIST-DECODING

This section explains the decoding process used in PCNN in more detail, and also provides a gentle introduction to polar codes.

**Simple Introduction to Polar Codes.**  We now describe polar codes and their decoding process. For ease of introduction we assume that the code dimension $\mathsf{cdim} = 2^t$ for some integer $t$, though this can be relaxed to any code dimension using code shortening/puncturing techniques (see, e.g., Zhang et al. (2014); Wang & Liu (2014); Saber & Marsland (2015); Niu & Chen (2012)). A polar code of rate $(\mathsf{cdim}, \mathsf{nbit})$ is defined by a mask $\mathbf{r} = (r_0, \cdots, r_{\mathsf{cdim}-1})$, which is a $\mathsf{cdim}$-dimensional binary vector in which exactly $\mathsf{nbit}$ entries equal 1. To encode a message word $\mathbf{m} = (m_0, \cdots, m_{\mathsf{nbit}-1})$ of $\mathsf{nbit}$ bits, one performs the following actions:

1. Create the binary pre-coded word $\mathbf{e} = (e_0, \cdots, e_{\mathsf{cdim}-1})$, such that

$$e_i := \begin{cases} 0 & r_i = 0 \\ m_j & i \text{ is the } j\text{'th nonzero coordinate in } \mathbf{r} \end{cases}$$

2. Apply the *polar transform* $f$ to $\mathbf{e}$ to obtain the codeword $\mathbf{c} = (c_0, \cdots, c_{\mathsf{cdim}-1})$.

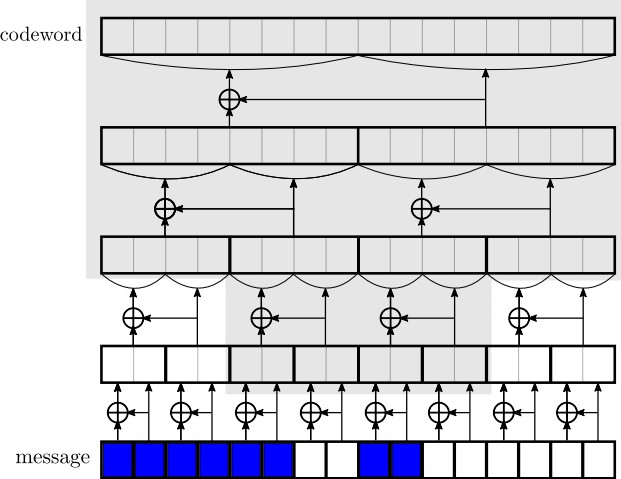

*Figure 5: Illustration of the decoding tree for polar code.*

The polar transform $f : \{0,1\}^{\mathsf{cdim}} \to \{0,1\}^{\mathsf{cdim}}$ in the above process can be described succinctly in the following way: if $\mathbf{c} = f(\mathbf{e})$ where for every index $i \in [\mathsf{cdim}]$, let $i = (\beta_0\beta_1\beta_2\ldots\beta_{t-1})$ be the binary representation of $i$. Define $L(i) := \{l | \beta_l = 1\}$ (the bits in $i$'s representation which are equal to 1). The polar transform is defined such that

$$c_i := \bigoplus_{j|L(j)\subseteq L(i)} e_j. \tag{1}$$

The above encoding process maps a $\mathsf{nbit}$-bit message word into a $\mathsf{cdim}$-bit code word, and can be shown to be linear. Thus, the image $C$ of this encoding is a $\mathsf{nbit}$-dimensional subspace of $\{0,1\}^{\mathsf{cdim}}$.

**Code Mask Generation.**    The correct choice of mask $\mathbf{r}$ is crucial for the error-correcting properties of the polar code. A good mask depends on the structure of the noise one aims to correct using the code. To generate a mask, we use an iterative process called genie-aided generation (Arikan, 2009): in this process, one subjects a codeword to the expected noise channel and studies which indices would be best for placing data bits (and which indices should be frozen). Since the points we aim to decode using PCNN are given in binary form, we chose to generate our masks using noise from a Binary Symmetric Channel (BSC), i.e., random bit flips.

The process for generating a mask for a pair $(\mathsf{cdim}, \mathsf{nbit})$ is only performed once, and is quite cheap computationally (in our case, involves decoding $\approx 10^7$ points). Moreover, since the mask is data-independent, masks can be reused globally across projects.

**(List) Decoding.**    Together with the introduction of polar codes, Arikan (2009) introduced the first decoding algorithm for polar codes, based on *successive cancelation* (SC). This algorithm is very efficient, and runs in time $O(\mathsf{cdim}\log\mathsf{cdim})$, i.e., nearly linear in the dimension of the codeword.

Tal & Vardy (2015) gave the first *list-decoding* algorithm, which returns $\mathsf{nprb}$ candidates for the closest codewords to the query, with time complexity $O(\mathsf{nprb} \cdot \mathsf{cdim}\log\mathsf{cdim})$ (i.e., the list size contributes linearly to running time). This list-decoding algorithm was made more computationally efficient by Hashemi et al. (2016), through pruning branches in the decoding tree; this algorithm achieves an improved decoding complexity of $O(\mathsf{nprb} \cdot \mathsf{cdim}\log\mathsf{nbit})$. This improved algorithm is the algorithm we use for PCNN. In our implementation, we use a heavily-modified version of the python-polar-coding library (https://github.com/fr0mhell/python-polar-coding), distributed under the MIT license.

To give some intuition for these decoding algorithms, Figure 5 shows the binary decoding tree used for encoding and decoding a polar code in which $\mathsf{cdim} = 16$ and $\mathsf{nbit} = 8$. The leaves of the tree represent the precoded word, where leaves corresponding to frozen coordinates in blue contain zeros and the remaining (unfrozen) leaves contain message bits. Each leaf in this tree contains one

message bit, except for the blue coordinates which are frozen (i.e., the mask value there is 0) and thus contain zeros. The root of this tree contains cdim bits, and represents the codeword. Each of the $2^{(\log \mathsf{cdim})-h}$ internal nodes of height $h$ in this tree contains an array of $2^h$ bits, which is calculated from the arrays of its two children. Roughly speaking, the SC decoding algorithm of Arikan (2009) performs a DFS traversal of this tree, filling all bit arrays. The time complexity is determined by the number of bits in those arrays, which is $O(\mathsf{cdim} \log \mathsf{cdim})$. The list decoding algorithm of Tal & Vardy (2015) also performs a DFS traversal of this tree, but maintains the nprb best results seen so far, which costs time $O(\mathsf{nprb} \cdot \mathsf{cdim} \log \mathsf{cdim})$. Finally, the simplified list decoding algorithm of Hashemi et al. (2016) prunes those nodes in the tree whose leaves are either all frozen or all unfrozen[4]; the remaining nodes are inside the gray outline. Note that for low-rate codes (i.e., nbit $\ll$ cdim) such as those used in PCNN, pruning such nodes yields a significant performance gain.

Note that these decoding and list-decoding algorithms are not exact, and sometimes return suboptimal results. However, we mitigate this in PCNN by increasing the list size of the algorithm beyond the desired list size. That is, to obtain the closest nprb codewords, we would use the algorithm of Hashemi et al. (2016) with list size $f(\mathsf{nprb}) \geq \mathsf{nprb}$, and only take the nprb best results. We have empirically found the following rule to nearly perfectly recover the closest codewords to the query:

$$f(\mathsf{nprb}) := \begin{cases} 16 & \mathsf{nprb} = 1 \\ 32 & 1 < \mathsf{nprb} \leq 16 \\ 2\mathsf{nprb} & 16 < \mathsf{nprb} \leq 256 \\ \mathsf{nprb} & \mathsf{nprb} > 256 \end{cases} \tag{2}$$

This rule is thus used in PCNN.

**Extraction of Cluster IDs.** In PCNN, after list-decoding a dataset point (during index creation) or a query (upon receiving one), we obtain the nprb closest codewords, each representing a cluster. While we could use the codewords themselves as cluster identifiers, this would be inefficient in terms of memory and storage, as each codeword has cdim bits and there are only $2^{\mathsf{nbit}}$ such codewords. Instead, we would like to extract from each codeword a nbit-bit cluster identifier which identifies the codeword uniquely. Given a codeword $\mathbf{c} = (c_0, \cdots, c_{\mathsf{cdim}})$, we extract the nbit-bit cluster identifier $x(\mathbf{c})$ through applying the code mask $\mathbf{r}$ to $\mathbf{c}$:

$$x(\mathbf{c}) := (c_i)_{i \mid r_i = 1} \tag{3}$$

We provide a proof that these identifiers are indeed unique.

**Proposition 1.** *Let $C$ be a $[cdim, nbit]$ polar code, $\mathbf{r}$ be its mask, and let $x$ be defined as in Equation* (3)*. Let $\mathbf{c}_1, \mathbf{c}_2 \in C$ be two codewords. Then,*

$$\mathbf{c}_1 = \mathbf{c}_2 \iff x(\mathbf{c}_1) = x(\mathbf{c}_2)$$

*Proof.* The left-implies-right direction is trivial (the cluster ID of a codeword is contained in the codeword), it remains to show the other direction. Assume that $\mathbf{c}_1 \neq \mathbf{c}_2$. Since the encoding process of polar code, as given in Equation (1), is injective, the distinct codewords $\mathbf{c}_1, \mathbf{c}_2$ were generated from two distinct nbit-bit message words $\mathbf{m}_1 = (m_0^1, \cdots, m_{\mathsf{nbit}}^1)$ and $\mathbf{m}_2 = (m_0^2, \cdots, m_{\mathsf{nbit}}^2)$. Moreover, each bit in each codeword is a linear combination (i.e., xor) of some subset of its message word. Denoting by $f$ the encoding, and defining $\mathbf{a}_1 := x(\mathbf{c}_1)$ and $\mathbf{a}_2 = x(\mathbf{c}_2)$, it holds that $\mathbf{a}_1, \mathbf{a}_2$ are created from $\mathbf{m}_1, \mathbf{m}_2$ through the linear transform $x \circ f$.

If we show that the linear map $g := x \circ f$ is injective, the proof is complete, as

$$\mathbf{c}_1 \neq \mathbf{c}_2 \implies \mathbf{m}_1 \neq \mathbf{m}_2 \implies \mathbf{a}_1 \neq \mathbf{a}_2$$

**Claim:** The linear map $g$ is injective.

We prove this claim through claiming that the image of $g$ (denoted $\mathrm{im}(g)$), which is contained in $\{0, 1\}^k$, has full dimension (i.e., equal nbit). Indeed, if this is not the case, then the perpendicular

---

[4]More accurately, the list-decoding algorithm of Hashemi et al. (2016) also prunes nodes with only a single unfrozen leaf (repetition nodes).

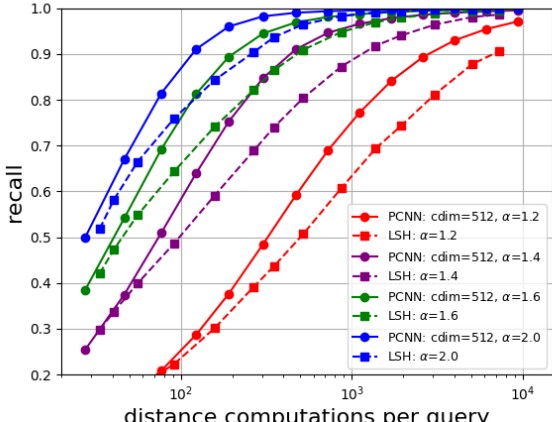

*Figure 6: Comparison of PCNN and LSH for varying Approximation Factor $\alpha \in \{1.2, 1.4, 1.6, 2.0\}$ on YandexDeep dataset (with sizenn $= 1$, using nbit $= 28$ and hyperplane LSH).*

space $\text{im}(g)^{\perp}$ contains a nonzero word - equivalently, there exists a subset $\emptyset \neq S \subseteq [\text{nbit}]$ such that

$$\forall \mathbf{a} = (a_0, \cdots, a_{\text{nbit}-1}) \in \text{im}(g) : \bigoplus_{j \in S} a_i = 0. \tag{4}$$

Suppose, for contradiction, that there exists such a nonempty set $S$. Now, for every $j \in [\text{nbit}]$ define $i_j$ to be the $j$'th one-valued bit in the mask $\mathbf{r}$.

For an index $i \in [\text{cdim}]$, denote by $L(i)$ the set of one-valued bits in $i$'s binary representation (as in the definition of polar codes in Equation (1)).

Now, fix $j \in S$ to be some index such that its location in the codeword is minimal according to $L$, i.e., $\forall j' \in S\backslash\{j\} : L(i_j) \nsubseteq L(i_{j'})$. Now, from the definition of $S$, it must be that $c_{i_j} = \bigoplus_{j' \in S\backslash\{j\}} c_{i_{j'}}$. However, recalling that for every index $i$ it holds that $c_i = \bigoplus_{i'|L(i) \subseteq L(i')} e_{i'}$, we have that $e_{i_j}$ goes into the xor of $c_{i_j}$, but not into the xor of $c_{i_{j'}}$ for any $j' \in S\backslash j$ (this uses the minimality w.r.t. $L$). But since $e_{i_j}$ can take on both zero and one, Equation (4) cannot hold for every cluster ID $\mathbf{a}$. This completes the proof. □

## B  ADDITIONAL EXPERIMENTS

### B.1  CHOICE OF APPROXIMATION FACTOR

The performance gains are present for any choice of approximation factor $\alpha$. In Figure 6, PCNN (solid curves) and LSH (dashed curves) are compared on *YandexDeep* dataset for $\alpha \in \{1.2, 1.4, 1.6, 2.0\}$. It is observed that PCNN outperforms LSH on every choice of approximation factor $\alpha$.

### B.2  CHOICE OF NEAREST NEIGHBOR SIZE

The performance gains are consistent for multiple choices of sizenn, the number of nearest-neighbors to output. Figure 7 depicts a comparison between PCNN (solid curves) and LSH (dashed curves) on *YandexDeep* dataset for sizenn $\in \{1, 10, 50\}$. It is observed that PCNN outperforms LSH for any choice of sizenn.

### B.3  ROBUSTNESS TO EMBEDDING RANDOMNESS

The performance of LSH can be greatly impacted by the choice of random embedding. We conjecture that this is due to the low number of embedding bits; thus, it is reasonable to assume that this variance in performance would be lower for PCNN, as it uses more embedding bits. To test

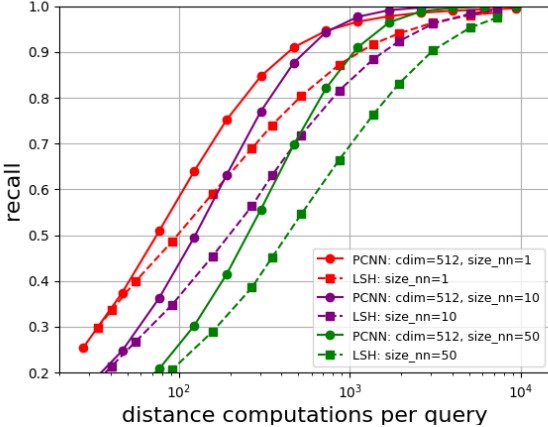

*Figure 7: Comparison of PCNN and LSH for varying Number of Neighbors* **sizenn** $\in \{1, 10, 50\}$ *on YandexDeep (with* $\alpha = 1.4$*, using* **nbit** $= 28$ *and hyperplane LSH).*

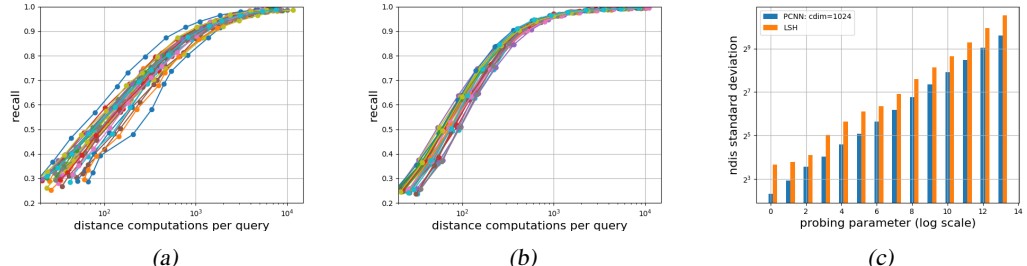

*Figure 8: Variance comparison for 30 different random seeds on YandexDeep (with* **sizenn** $= 1$*,* $\alpha = 1.4$ *and using* **nbit** $= 28$ *and hyperplane LSH) for (a) CHC, (b) PCNN with* **cdim** $= 1024$*. Sub-Figure (c) shows the standard deviations from (a) and (b) per choice of* **nprb***.*

this conjecture, we ran both LSH and PCNN with 30 different random seeds. We considered the *YandexDeep* dataset, **sizenn** $= 1$ and $\alpha = 1.4$, and ran both LSH and PCNN with **nbit** $= 28$. For PCNN, we chose **cdim** $= 1024$. The resulting measurement can be seen in Figure 8; Figures 8a and 8b show the performance for LSH and PCNN respectively, while Figure 8c shows the standard deviations of both algorithms for every choice of **nprb**. A reduced variance in the performance of PCNN versus LSH is clearly observed.

## B.4 CONVEX FRONTIER

Our previous experiments averaged performance over the random seed of the algorithm. In standard usage, where one chooses an arbitrary random seed, this method seems reasonable. However, one could imagine trying to optimize for the best random seed (which, as mentioned above, would greatly impact the performance of CHC). A natural question would be whether the higher variance of CHC compared to PCNN would make the best seed choice for CHC better than the best seed choice for PCNN. We answer this in the negative: PCNN still outperforms CHC in this case.

To consider this seed-optimization regime, we repeat previous experiments where instead of averaging over seeds, we take only the vertices of the convex hull of the algorithm's results (over all seeds). We then prune the vertex set by taking its Pareto frontier (i.e., a subset of points such that no point in the subset is worse in both **recall** and **ndis** than another vertex). Figure 9 shows the choice of this convex frontier from the runs of an algorithm with 30 different seeds (in this figure, the frontier is dominated by a single seed, but this need not always be the case). Figure 10 compares the convex frontiers of the various algorithms.

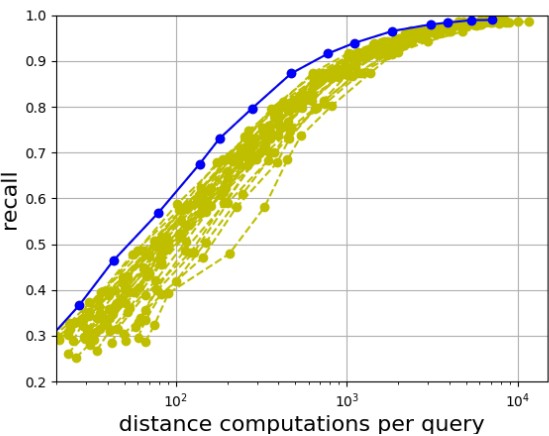

*Figure 9: Illustration of a convex frontier of CHC (solid blue curve) that corresponds to 30 curves (dashed yellow curves) obtained by using different seeds as used in the case of Figure 8a.*

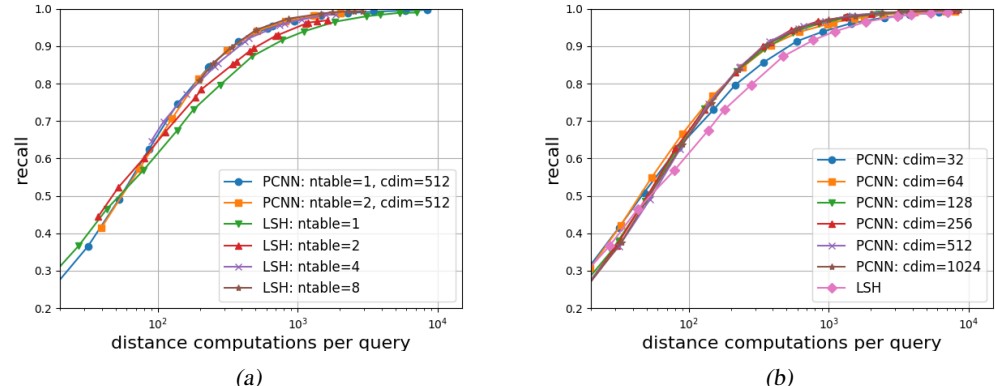

*Figure 10: Reproducing the results of Figures 3a and 4a using convex-frontier instead of seed averaging.*

### B.5 APPROXIMATE HYPERPLANE LSH

As previously stated, the results above regarding the choice of cdim seem to indicate that it should be roughly on par with the original real dimension $d$. However, in hyperplane LSH choosing $\mathsf{cdim} = d$ implies multiplication by a $d \times d$ matrix with normally-distributed entries upon embedding a vector, which would take $O(d^2)$ time. However, there exists an efficient alternative for this multiplication; this alternative involves repeatedly flipping entry signs at random, then applying a Hadamard transform. After a constant number of iterations, this process has been seen to approximate the original matrix multiplication, while taking only $O(d \log d)$ time (Andoni et al., 2015). (Such approximate embeddings are based on the fast Johnson-Linderstrauss transform of Ailon & Chazelle (2009), and were also considered by, e.g., Dasgupta et al. (2011).)

We test this method for PCNN, and observe that 4 iterations of this process (sign flip + Hadamard transform) are sufficient for identical performance to hyperplane LSH. Figure 11 shows the results of this experiment.

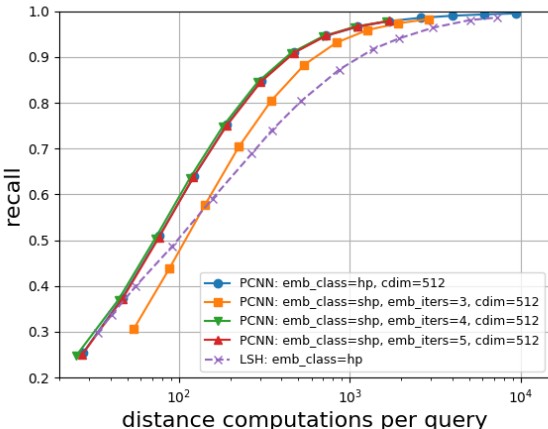

*Figure 11: Comparison of PCNN with hyperplane embedding vs. Hadamard-based embedding (with hyperplane LSH for comparison). The dataset is YandexDeep, with sizenn = 1 and $\alpha = 1.4$.*

### B.6 Autoencoder LSH

PCNN is a general method which can encapsulate any LSH function to obtain an algorithm for nearest-neighbor search. Thus far, we have focused on hyperplane LSH as a concrete, classic example for such an LSH function. However, other LSH functions can also be used by PCNN; this subsection focuses on exploring one such LSH function, namely the data-dependent autoencoder LSH introduced by Tissier et al. (2019).

In this subsection, we explore the performance gains of PCNN over CHC, where both techniques encapsulate the autoencoder LSH method. We implemented the autoencoder architecture of Tissier et al. (2019) in pytorch lightning, and trained it on the *YandexDeep* dataset. For this, we used a regularization parameter $\lambda_{reg} = 1$ (as defined by Tissier et al. (2019)), and ran the Adam optimizer with a learning rate of 0.001 and a batch size of 128. For every value of cdim, we trained an autoencoder in this way which has a representation layer of size cdim.

First, we compare CHC with PCNN with various choices of cdim; we do this for the *YandexDeep* dataset, similar to presented performance gains in Section 4.4 for hyperplane LSH. The choice of problem parameters is the same as for hyperplane LSH, i.e., sizenn = 1 and $\alpha = 1.4$, as is the parameter nbit = 28. The results, given in Fig. 12, show a dramatic difference. This is since a larger representation allows for a better autoencoder, and PCNN provides access to these larger representations.

Next, we repeat the experiment of Section 4.4 for Multi-Table LSH with autoencoder LSH. This experiment is again on the *YandexDeep* dataset, with problem parameters sizenn = 1 and $\alpha = 1.4$, as well as nbit = 28. The results are given in Fig. 13, and show similar results to those seen for hyperplane LSH. The main difference is that using multiple tables improves the performance of PCNN even in large cdim (i.e., 512); we attribute this to the slower plateau of performance w.r.t. cdim that the autoencoder LSH exhibits in comparison to hyperplane LSH.

## C Approximation Factor for Cosine Similarity

In the case of *YandexTTI* dataset we use cosine distance, defined as one minus the cosine similarity. While maximizing cosine similarity and minimizing cosine distance are equivalent, using an approximation factor $\alpha$ for cosine distance could yield little intuition regarding the minimal value of the corresponding cosine similarity. This is due to the fact that when emanating from approximation factor $\alpha \geq 1$ defined for cosine distance, the corresponding approximation factor for cosine similarity does not remain constant and depend on the value of the reference cosine similarity. Formally, let $cd_1$ and $cd_2$ denote two cosine distances, and let $cs_1$ and $cs_2$ denote the corresponding cosine similarities, respectively. It holds that $cd_2 \leq \alpha \cdot cd_1$ if and only if $cs_2 \geq (\alpha - (\alpha - 1)/cs_1)) \cdot cs_1$. Table 2 shows the effect of choosing different $\alpha$ for different cosine similarity values. The values in

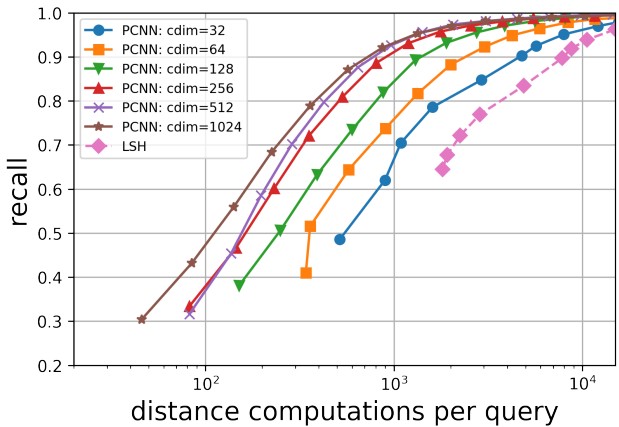

Figure 12: *recall/ndis* performance comparison for **sizenn** $= 1, \alpha = 1.4$ for *YandexDeep (using* nbit $= 28$ *and autoencoder LSH).*

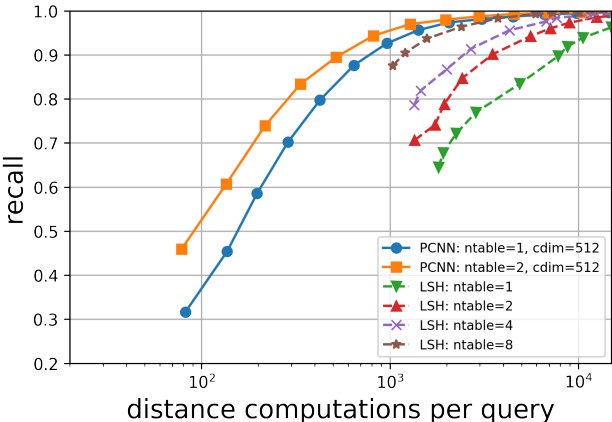

Figure 13: *recall/ndis* performance comparison for **sizenn** $= 1, \alpha = 1.4$ for *YandexDeep (using* nbit $= 28$ *and autoencoder LSH).*

the left column correspond to values of $cs_1$ (e.g., reference cosine similarity values) while the values in the body of the table correspond to values of $cs_2$ (minimum allowed cosine similarity value by approximation ratio) with different $\alpha$ per column. For example, if the optimal reference cosine similarity equals $0.97$, than approximation factor $\alpha = 1.5$ for cosine distance implies that any cosine similarity of at least $0.955$ is valid (that is, an effective approximation ratio of $0.984$ w.r.t. cosine similarity).

Table 2: *Examples of changes to cosine similarity through approximation factor $\alpha$ for cosine distance.*

| $cs_1$ \ $\alpha$ | 1.2 | 1.5 | 2.0 | 2.5 | 3.0 |
|---|---|---|---|---|---|
| 0.7 | 0.64 | 0.55 | 0.4 | 0.25 | 0.1 |
| 0.8 | 0.76 | 0.7 | 0.6 | 0.5 | 0.4 |
| 0.9 | 0.88 | 0.85 | 0.8 | 0.75 | 0.7 |
| 0.95 | 0.94 | 0.925 | 0.9 | 0.875 | 0.85 |
| 0.97 | 0.964 | 0.955 | 0.94 | 0.925 | 0.91 |

# D  COMPLEXITY DISCUSSION

**Time complexity of PCNN versus CHC.** In both the hyperplane and autoencoder LSH methods with nbit-bit hashcodes, CHC would have a preprocessing time complexity of $n \cdot O(\text{embedding cost} + \text{nbit})$ and a query complexity of $O(\text{embedding cost} + \text{nprb} \cdot \text{nbit})$. (Not included in this is the cost of training the autoencoder model, which varies depending on the training parameters.) In order to compare PCNN to CHC, we must pick a regime for cdim; for hyperplane LSH, for example, our experiments show that picking $\text{cdim} \approx d$ optimizes performance, and thus in the following comparison we assume $\text{cdim} = d$. In addition, we should consider the cost of binary embedding: embedding to a dimension of $d_{\text{bin}}$ takes $O(d_{\text{bin}} \cdot d)$ time in standard hyperplane embedding. We also consider the complexity of an approximate embedding based on Hadamard transform and random sign flips, identical to that in Andoni et al. (2015); this approximate embedding takes $O((d_{bin} + d) \log(d_{bin} + d))$ time.

Table 3 summarizes the running time complexities of both PCNN and CHC. Overall, for both PCNN and CHC, the cluster ID extraction upon query is not significant in terms of running time; in both cases, the determining component in running time is the distance comparisons between the query and the dataset points (ndis, as defined in Section 4.2).

Table 3: Comparison of the time complexity of PCNN and CHC. Time complexity is compared for preprocessing and for extraction of nprb cluster IDs (CIDs) to probe upon a query. (Note that this table does not refer to the main cost of querying, which is searching within clusters; this is evaluated empirically in Section 4.)

|  |  | without embedding | w. hyperplane embedding | w. approx. embedding |
|---|---|---|---|---|
| PCNN | preprocess | $O(n \cdot d \cdot \log d)$ | $O(n \cdot d^2)$ | $O(n \cdot d \cdot \log d)$ |
|  | extract CIDs | $O(\text{nprb} \cdot d \cdot \log d)$ | $O(d^2 + \text{nprb} \cdot d \cdot \log d)$ | $O(\text{nprb} \cdot d \cdot \log d)$ |
| CHC | preprocess | $O(n \cdot \text{nbit})$ | $O(n \cdot d \cdot \text{nbit})$ | $O(n \cdot d \cdot \log d)$ |
|  | extract CIDs | $O(\text{nprb} \cdot \text{nbit})$ | $O(d \cdot \text{nbit} + \text{nprb} \cdot \text{nbit})$ | $O(d \cdot \log d + \text{nprb} \cdot \text{nbit})$ |

