# OpenReview forum: "Approximate Nearest Neighbor Search through Modern Error-Correcting Codes"
_ICLR.cc/2023/Conference — ICLR 2023 poster_

### Official Review · Reviewer_8eNv · 2022-10-23

**Confidence:** 3
**Correctness:** 3
**Technical Novelty And Significance:** 3
**Empirical Novelty And Significance:** 2
**Recommendation:** 6

**Clarity, Quality, Novelty And Reproducibility:**

This paper is well-written, and the figure makes the techniques easy to follow.

A minor issue with the clarity is that readers might not be familiar with the Polar code and the list-decoding methods. They might shorten the experimental setup (e.g., simplify the datasets and provide more details in the supplementary material) to make room for moving the background of the Polar code and the list-decoding methods to the main paper.

**Strength And Weaknesses:**

The paper has the following strengths:

S1. This paper is well-motivated. Vanilla methods often embed the original data points into a low-dimensional latent space, which usually distorts the distance for many pairs of data points. Thus, considering a high-dimensional latent space might be promising.

S2. The experimental analysis is extensive. The experimental results demonstrate the superiority of the proposed approach compared with CHC. Other interesting experiments related to the impact of $\alpha$ and \textsf{sizenn}, robustness and extension to different LSH functions are also included in the supplementary material.

The paper has the following weaknesses:

W1. Even though I like the motivation, the authors only directly leverage the polar codes and the list decoding method for multi-probing, which makes the novelty somehow limited.

W2. They indeed embed the data point in a high-dimensional latent space (i.e., \textsf{cdim} bits), but the closest clusters they aim to find for the query are represented in a low-dimensional space (i.e., \textsf{nbit} bits). Can you provide more insights into the benefits of the high-dimensional latent space?

W3. I think the \textsf{ndis} is a good measurement if we want to ignore the benefits from the implementation level; it could be better if the author could also contain the embedding cost. For example, for the hyperplane LSH by Charikar (2002), as both distance cost and a single LSH function cost are O(d), each LSH function can be included and considered once \textsf{ndis}.

W4. The experiments provide many results about different \textsf{cdim} values. As \textsf{nbit} is also a vital parameter for PCNN, Can the authors also provide some more results about the impacts of different \textsf{nbit} values?

W5. They provide pre-processing time and query time complexities for PCNN. To make a complete analysis. It will be nice to provide space analysis. It would also be nice to extend the experimental analysis with a scalability study to examine how the \textsf{ndis} evolves as the size of the input data increases.



**Summary Of The Paper:**

In this paper, the authors investigate a problem in clustering by locality-sensitive hashing for similarity search. They introduce a new method called the Polar Code Nearest Neighbor (PCNN) method that uses the polar codes to maintain a number of clusters in a high-dimensional embedding space. By utilizing the list-decoding method, they propose a multi-probe scheme for PCNN to efficiently determine the closest clusters to the query. Extensive experiments confirm the superior performance of PCNN over the classic hash clustering (CHC).


**Summary Of The Review:**

In summary, the authors did a good job by leveraging the Polar code and the list-decoding method for similarity search, but I still have some concerns about its effectiveness and some issues in the experiments. Thus, I initially give a borderline acceptance.

---

> ### Author Response · Authors · 2022-11-15
> **Response to Reviewer 8eNv**
>
> We thank the reviewer for the time and effort invested into the review of our manuscript.
>
> Regarding W2: First, note that while the clusters in PCNN form a low-dimensional subspace, each point has a high-dimensional representation which is used in the algorithm. The use of good ECCs provides an informative high-dimensional representation rather than naive extensions of low-dimension representations to high-dimension.
>
> Here is some intuition as to why a high-dimensional representation is useful in multiprobe: in multiprobe, one aims to probe the closest clusters to the query. However, in CHC, standard multiprobe would probe the closest clusters to the \emph{closest cluster} to the query. (This is since the query q is first embedded to a latent space point b which is its closest cluster, and b is then perturbed for additional clusters.) This is not desired, as the latent space in CHC is (necessarily) low-dimensional, resulting in distance distortion. Meanwhile, in PCNN, the latent space point b preserves the distances of the query q as the space is high-dimensional; thus, when searching for the closest clusters to b, we are in fact searching for the closest clusters to q, as desired.
>
> Regarding W3: Note that, like CHC, PCNN only applies the LSH function once upon each query. Thus, this does not affect the ndis comparison. In terms of complexity, you may refer to Table 3; to summarize, good hyperplane embeddings can be generated in time quasi-linear in the dimension (e.g., see in addition Appendix B.5).
>
> Regarding W4: In general, a higher nbit yields more clusters and thus a more precise clustering. However, if the number of clusters is so large that the average cluster is empty, the cost of probing the clusters starts to dominate, overshadowing ndis as a dominant cost component. To avoid this, we chose nbit to be such that the average probed cluster contains a small, constant number of dataset points; still number of cluster is linear in the size of the dataset D. We chose not to vary nbit in this paper as the criteria for choosing this parameter are identical for CHC and PCNN.
>
> Regarding W5: A single table in PCNN takes exactly the same storage as a single table in CHC, fixing the total number of clusters (i.e., 2^nbit). This fact is referred to in “Multi-Table LSH” in Subsection 4.4. This makes use of a concise representation of polar code words as information words of length nbit (see, e.g., cluster ID extraction in Figure 2).

---

### Official Review · Reviewer_FYov · 2022-10-24

**Confidence:** 4
**Correctness:** 4
**Technical Novelty And Significance:** 4
**Empirical Novelty And Significance:** 4
**Recommendation:** 8

**Clarity, Quality, Novelty And Reproducibility:**

While the paper says in part that it uses a standard (freely available) python library - it does say "heavily modified" and it isn't clear to me how much of the three algorithms that make up the overall scheme, are "covered" but the library - the placing of the aforementioned comment, in the paper, suggests only for algorithm 3. However, they declare they will make the code available. That would *suggest* that the reproducibility should be somewhat assured.

The novelty appears to be clearly in the recognition that polar codes could be used in ANNS. It seems to me, from my standpoint, that unless there is missed prior art in that respect, then the novelty is sufficient. I am not all that clear on how much of an inspirational jump it is to make the connection but at the moment I am inclined to believe it is significant. (BTW on the "forced choice" questions below I can only choose "The contributions are significant, and do not exist in prior works." because all other options seem too strong in the implied deficiencies they offer). Somewhat ditto on empirical support - even though this is one area the paper could be improved, clearly.

Overall, the clarity of the paper and the quality seem good to me. That said, I find the main part a bit slow and repetitive and suspect it would be improved by making it tighter and space saving so that more of the appendices could be fitted into the main paper.

**Strength And Weaknesses:**

The strengths are that the proposal seems sound - using polar codes seems like a good thing to do and this is supported by the experiments. The paper is well written in most ways. So I believe this is a solid paper with a solid contribution and written at least welklk enough for acceptance.

As per the next section - the novelty *whilst probably enough* doesn't seem that inspirational and that insightful *on my reading* of the *current manuscript*. In that regard, perhaps the hypothesis that the cdim=d is a good choice (mentioned in the footnote 2 and elsewhere) might be the more interesting aspect to the work from my perspective - and it's a pity there isn't more light on this. That said, the paper is solid enough to be a really strong contender for publication even in the current form.

The range of data sets seems limited. I don't want to make strong point here - however it would be better to demonstrate performance on a wider variety of database types and hence potential applications - but I am mindful of the time cost and decline pressures.



**Summary Of The Paper:**

Takes the idea of polar codes (and polar coding) to develop an approximate nearest neighbour scheme with some efficiency.
Investigates the application of that to search in three databases - all three containing SIFT visual descriptors but one containing text as well.
The claims and results tend to show that the method produces better recall than the competitors (principally variants of hyperplane LSH with various numbers of multiple tables; although another version of LSH using a (data dependent) hyperplane chosen using the method of Tissier et. al is also briefly investigated).
This better performance on recall is despite using less than the sort(D) distance computations for a dataset of size D, generally required for the comparators.

**Summary Of The Review:**

It seems to me the gains are sufficient to be of interest to people (working in or reliant on ANNS). It seems to me the idea is sound, and if not already published, should be. The paper is written well enough. These three things - to me - are completing for acceptance.

---

> ### Author Response · Authors · 2022-11-15
> **Response to Reviewer FYov**
>
> We thank the reviewer for the time and effort invested into the review of our manuscript.
>
> Regarding datasets: note that only one of the datasets contains SIFT descriptors (BIGANN). The YandexDeep dataset contains learned DNN embeddings of images, while the YandexTTI dataset contains joint DNN embeddings of both text and images. We chose to focus on these 3 types of datasets as they are representative for settings of billion-scale datasets of dense real vectors.
>
> Regarding reproducibility: The python code we will provide covers all parts of the PCNN algorithm (Algorithms 1-3). We implement the list decoder of Hashemi et al. through modifying an existing polar code library; our modifications include fixing various bugs and improving running-time complexity to be quasi-linear in cdim.
> The code also contains scripts for processing of datasets, running provided evaluations, and generating the figures. We added a paragraph about code reproducibility to the submission draft.
>
> Regarding the hypothesis that cdim=d is a good choice: In hyperplane LSH, the expected relative Hamming distance of the hash-codes is equal to the relative angular distance between the original real points (Charikar, 2002). The deviations from the expectation are very significant when the number of bits is low (See gap paragraph before “Our contributions” on Section 1). In the case of the experimented datasets, we empirically observe reaching a plateau in the recall performance when exceeding cdim=d (Figure 3). We assume that for these cases cdim=d provides a granular precise enough estimation for the distance between the original real points and using more bits (cdim>d) does not provide additional information. Note that this is observed in the experimented settings of dense real datasets that are likely to have almost no statistical redundancy. If the dataset dimension d is higher and exhibits statistical redundancy, then this plateau may be reached with cdim<d (maybe cdim \simeq empirical entropy of D). We may conjecture that cdim=d is sufficient but not necessary in the general case (see “Performance Gains” in Section 4.4).

---

### Official Review · Reviewer_sMa7 · 2022-10-25

**Confidence:** 4
**Clarity, Quality, Novelty And Reproducibility:** The proposed PCNN algorithm is easy t…
**Correctness:** 3
**Technical Novelty And Significance:** 4
**Empirical Novelty And Significance:** 3
**Recommendation:** 6

**Strength And Weaknesses:**

Strength:
1. The combination of LSH and error-correction code is novel.

2. This paper is well-organized. Figure 1 is illustrative for readers to understand the functionality of PCNN

3. Large ANNS datasets are used for empirical evaluation, making a fair comparison of PCNN with LSH

Weakness:
1. Stronger arguments is required for the motivation of applying  error-correction code in ANNS.

2. In the evaluation, the comparison of PCNN with product quantization approaches is missing.

3. For binary datasets, MinHash algorithms should also be considered as a baseline.



**Summary Of The Paper:**

This paper studies the approximate nearest neighbor search problem.  Specifically, this paper proposes a Polar Code Nearest-Neighbor (PCNN) algorithm. This algorithm uses error-correcting codes to build clusters in the latent space in high dimensionality.  Compared to classical LSH clustering, PCNN uses few hash tables to achieve the same performance. As a result, PCNN is memory efficient compared to LSH.

**Summary Of The Review:**

This paper introduces the error-correction code techniques in the field of ANNS, This combination is novel since it opens a new direction for ANNS. However, both the motivation and empirical evaluation can be further improved for better quality.

Questions
1. Is there any justification for why error-correction code should be applied in ANNS? Why is error-correction code a better way to manage LSH functions?

2. Is it possible to provide theoretical guarantees on the clustering quality of PCNN?

3. In ANNS evaluation, it is inevitable to compare with state-of-the-art graph/quantization approaches for efficiency-accuracy tradeoffs. In this setting, LSH-based approaches may not be the best choice for dense vectors. I would recommend a study in high dimensional and sparse regime with a comparison to Minhash

4. Can we provide privacy guarantees for PCNN?

---

> ### Author Response · Authors · 2022-11-15
> **Response to Reviewer sMa7**
>
> We thank the reviewer for the time and effort invested into the review of our manuscript.
>
> Regarding W1 and Q1: In essence, the field of error-correcting codes focuses on very efficient techniques for solving the nearest-neighbor problem on specially-crafted datasets (error-correcting codes). In clustering methods for ANNS, we first cluster the dataset, and then solve ANNS for the set of clusters (rather than for the original dataset). Choosing the set of clusters to be an ECC allows us to use these techniques. Furthermore, it allows the representation of both clusters and queries in high-dimension (denoted in the paper by cdim) which results in lower distortion of the distances between two real vectors when projected from the real high-dimension (d) space of dataset D, while maintaining a smaller number of clusters (2^nbit).
> We provide these arguments for the use of ECC in Section 1 within the first 2 paragraphs of “our contribution” subsection and first paragraph of Section 3 (The PCNN Algorithm).
>
> Regarding W2: Product quantization is used to speed up distance computations between points, at some accuracy cost. The method for computing pairwise distances is orthogonal to the novelty of PCNN, and thus product quantization can be used in both CHC and PCNN, and would arguably yield similar results.
>
> Regarding Q2: Theoretical analysis for this problem seems to be less appropriate, as implied from previous work:
> * Existing analyses of classic multiprobe techniques (e.g., CHC) have been practical rather than theoretical.
> * All existing analyses of list-decoding for polar codes have been practical, providing us with no theoretical guarantees for finding nearest codewords.
>
> Regarding W3 and Q3: MinHash is a fast method for approximating the Jaccard similarity of two sets over a finite number of elements. Indeed, it may be applied to approximate nearest neighbors search in settings of very high-dimension (e.g., >10^5) and very sparse datasets (e.g., <10^-3 fraction non-zeros entries). In our paper we focused on the setting of datasets with dense real/binary representations and use of Hamming distance to which MinHash is less applicable. Dealing with settings of very high dimensions and sparse datasets is an appealing direction for future work.
>
>
> Regarding Q4:
> We note that ECC transformation does not provide cryptographic strength for masking private data, however are amenable to distributed multiparty computing settings. Many aspects of PCNN remain to be studied, and this could be an interesting topic for future work.

---

### Official Review · Reviewer_96rD · 2022-11-04

**Confidence:** 2
**Correctness:** 4
**Technical Novelty And Significance:** 2
**Empirical Novelty And Significance:** 2
**Recommendation:** 3

**Clarity, Quality, Novelty And Reproducibility:**

The writeup is clear, but hard to evaluate lacking any concrete claims that the scheme generally works, or works under certain assumptions.

**Strength And Weaknesses:**

The paper does not prove any rigorous guarantees or bounds on the proposed scheme. It's validated solely empirically.

**Summary Of The Paper:**

The paper proposes a locality sensitive hashing scheme, applicable to approximate nearest neighbor search, using Polar codes as the sparse collection of hash values.

**Summary Of The Review:**

It's an interesting idea, but its validation is not convincing.

---

> ### Author Response · Authors · 2022-11-15
> **Response to Reviewer 96rD**
>
> We thank the reviewer for the time and effort invested into the review of our manuscript.

---

### Decision · Program_Chairs · 2023-01-20

**Decision:**

Accept: poster

**Justification For Why Not Higher Score:**

Nearest neighbor search is an important component of many real-world machine learning systems, so advances in its efficiency have potentially wide impact.  On the other hand, the paper's improvements are presented primarily over related baselines, so its efficacy in industrial-strength systems (cf SPANN/Faiss) remains to be seen.

**Justification For Why Not Lower Score:**

All reviewers consider the use of polar codes to be an interesting idea, and no reviewer provides any objection to the claim of novelty.


**Metareview: Summary, Strengths And Weaknesses:**

All reviewers consider the use of polar codes to be an interesting idea, and no reviewer provides any objection to the claim of novelty.

Two reviewers ask for further theoretical analysis.   The rebuttal argues that such an analysis is not available for nearby multiprobe techniques, so would be difficult here.  In AC/Reviewer discussion it was agreed that such analysis might well prove to be future work for these or other authors, and so its absence should not preclude publication of the paper.

The claim, in the rebuttal, that product quantization should be orthogonal to the novelty of this work, was considered plausible, but it would greatly improve the paper to include experiments that test that hypothesis.  Particuarly given the discussion of empricial vs theoretical contribution, it would be appropriate to empirically confirm this claim.

Other reviewers' questions were adequately answered in rebuttal.

**Note From Pc:**

if the above contains the word "oral" or "spotlight" please see: "oral" presentation means -> notable-top-5% and "spotlight" means -> notable-top-25%. As stated in our emails, we are disassociating presentation type from AC recommendations

**Summary Of Ac-Reviewer Meeting:**

The primary point of discussion at the meeting was the need for theoretical analysis.  As described above, reviewers all agreed that the contribution was sufficient given the current empirical analyses.

As a result of the meeting some further related work was noted; this of course is entirely optional for the authors to include.

- https://arxiv.org/pdf/1712.08558.pdf
- https://web.mit.edu/andoni/www/papers/cSquared.pdf
- https://www.cs.princeton.edu/courses/archive/spr05/cos598E/bib/kushilevitz.pdf